# Therapeutic Effects of a Novel Form of Biotin on Propionic Acid-Induced Autistic Features in Rats

**DOI:** 10.3390/nu14061280

**Published:** 2022-03-17

**Authors:** Kazim Sahin, Cemal Orhan, Serdar Karatoprak, Mehmet Tuzcu, Patrick Brice Defo Deeh, Ibrahim Hanifi Ozercan, Nurhan Sahin, Merve Yilmaz Bozoglan, Sarah Sylla, Sara Perez Ojalvo, James R. Komorowski

**Affiliations:** 1Department of Animal Nutrition, Faculty of Veterinary Medicine, Firat University, Elazig 23119, Turkey; corhan@firat.edu.tr (C.O.); nsahin@firat.edu.tr (N.S.); 2Department of Child and Adolescent Psychiatry, Mental Health Hospital of Elazig, Elazig 23119, Turkey; srdrkrtprk@hotmail.com; 3Division of Biology, Faculty of Science, Firat University, Elazig 23119, Turkey; mtuzcu@firat.edu.tr; 4Department of Animal Biology, Faculty of Science, University of Dschang, Dschang P.O. Box 96, Cameroon; deehdefo@yahoo.fr; 5Department of Pathology, Faculty of Medicine, Firat University, Elazig 23119, Turkey; ihozercan@firat.edu.tr; 6Department of Pharmacology, Faculty of Medicine, Firat University, Elazig 23119, Turkey; mybozoglan@firat.edu.tr; 7Scientific and Regulatory Affairs, JDS Therapeutics, LLC, Purchase, NY 10577, USA; ssylla@nutrition21.com (S.S.); sperezojalvo@nutrition21.com (S.P.O.); jkomorowski@jdstherapeutics.com (J.R.K.)

**Keywords:** magnesium biotinate, biotin, propionic acid, oxidative stress, neuroinflammation autism

## Abstract

Magnesium biotinate (MgB) is a novel biotin complex with superior absorption and anti-inflammatory effects in the brain than D-Biotin. This study aimed to investigate the impact of different doses of MgB on social behavior deficits, learning and memory alteration, and inflammatory markers in propionic acid (PPA)-exposed rats. In this case, 35 Wistar rats (3 weeks old) were distributed into five groups: 1, Control; 2, PPA treated group; 3, PPA+MgBI (10 mg, HED); 4, PPA+MgBII (100 mg, HED); 5, PPA+MgBIII (500 mg, HED). PPA was given subcutaneously at 500 mg/kg/day for five days, followed by MgB for two weeks. PPA-exposed rats showed poor sociability and a high level of anxiety-like behaviors and cognitive impairments (*p* < 0.001). In a dose-dependent manner, behavioral and learning-memory disorders were significantly improved by MgB supplementation (*p* < 0.05). PPA decreased both the numbers and the sizes of Purkinje cells in the cerebellum. However, MgB administration increased the sizes and the densities of Purkinje cells. MgB improved the brain and serum Mg, biotin, serotonin, and dopamine concentrations, as well as antioxidant enzymes (CAT, SOD, GPx, and GSH) (*p* < 0.05). In addition, MgB treatment significantly regulated the neurotoxicity-related cytokines and neurotransmission-related markers. For instance, MgB significantly decreased the expression level of TNF-α, IL-6, IL-17, CCL-3, CCL-5, and CXCL-16 in the brain, compared to the control group (*p* < 0.05). These data demonstrate that MgB may ameliorate dysfunctions in social behavior, learning and memory and reduce the oxidative stress and inflammation indexes of the brain in a rat model.

## 1. Introduction

Autism spectrum disorder (ASD) is a neurodevelopmental disorder characterized by persistent social communication and interaction deficits, limited repetitive behavioral characteristics, and interests [1]. ASD is observed more frequently nowadays and represents a severe public health problem. For instance, in the United States, the prevalence of ASD increased from 1/68 in 2012 to 1/59 in 2018 [2]. This increase has been attributed to the factors such as changes in diagnostic criteria, increased awareness of ASD, and availability of ASD services. [2]. The etiology of ASD is not well understood yet, and it is believed to occur as a result of genetic and environmental aspects affecting brain development [3]. In previous studies, microanatomic (reduced Purkinje cells in the cerebellum, decreased connectivity between the frontal gyrus and posterior regions, insula and the amygdala) and macroanatomic (increase in brain volume in childhood, increased volume in the frontal and temporal lobes, reduced cerebellar volume, volume alterations in the hippocampus and amygdala) differences were detected in the brain of patients with ASD [4,5]. Alterations in hippocampal structures which mediate emotion perception and regulation are related to ASD behaviors [6]. In addition, neurochemical changes associated with alterations in the serotonin and dopamine metabolisms have been found in individuals with ASD and propionic acid (PPA)-induced autism-like rats [7,8]. Moreover, several studies showed that ASD is associated with elevated lipid peroxidation, impaired methylation, and low antioxidant enzymes (glutathione, catalase, superoxide dismutase) involved in the defense system against reactive oxygen species (ROS) [9,10]. Concerning oxidative stress, proinflammatory markers including TNF-α, interleukins (IL-6, IL-17), and chemokines (CCL-3, CCL-5, and CXCL-16) have been involved in the etiology of ASD [11,12,13]. Indeed, overexpression of TNF-α, IL-6, and IL-17 genes observed in the brain of patients with autism seriously impaired the dendritic spine development and promoted autistic-like features [14]. Since chemokines (CCL-3, CCL-5, CXCL-16) regulate CNS inflammatory states or neurogenesis, their abnormal levels in the brain promote the pathogenesis of ASD [15,16]. Moreover, the abnormal level of pro-inflammatory markers is positively correlated with neuroplasticity-related markers such as BDNF and ICAM [17,18]. BDNF inhibits apoptosis and promotes neuronal reorganization, while ICAM improves tissue formation, maintenance, and function [19]. However, the exact mechanism is unknown.

The treatment of ASD, such as its etiology, is still unclear today. A pharmacological drug has not yet been created to treat core symptoms of autism effectively. Therefore, many different treatment options are being studied to treat ASD. However, the results of the studies are inconsistent. For example, some studies suggest that vitamin/mineral supplements may affect autism symptoms, while some suggest that these supplements have no effect [20,21]. Biotin is a vital cofactor for five known carboxylases involved in carbohydrate, lipid, and protein metabolism and gene regulation, particularly for genes involved in carbohydrate metabolism [22]. Biotin plays an essential role in multiple cellular processes, including histone modification, cell proliferation, DNA repair, and protein expression, glucokinase, holocarboxylase synthetase, and sodium-multivitamin transporter expression [23]. In biotin-deficient individuals, serious clinical abnormalities occur, including growth retardation neurological and dermatological disorders [24]. Genetic disorders such as biotinidase deficiency or holocarboxylase synthetase deficiency may also result in biotin deficiency [25]. Similarly, magnesium (Mg) shows a key role in several cellular processes such as oxidative phosphorylation, glycolysis, cellular respiration, and protein synthesis [26]. Mg deficiency causes several symptoms such as tetany, seizures, arrhythmia, and neuromuscular irritability. Low Mg levels or Mg deficiency may also contribute to hypocalcemia and is also involved in many age-related phenotypes, including sarcopenia and metabolic syndrome [27].

Studies on the effects of Mg and biotin in the etiology and treatment of ASD are limited. Colamaria et al. [28] reported that an infant suffering from progressive lethargy, sparse scalp hair, autistic-like behavior, myoclonus, and drug-resistant generalized seizures showed a dramatic improvement in all symptoms and signs after biotin therapy. Similarly, Benke et al. [29] reported nail growth, verbal expression skills, and school performance increased after biotin treatment in a child with autism, nail and hair growth deficiency. On the contrary, in another report, in a patient with partial biotinidase deficiency and ASD, it was found that biotin treatment only relieved Candida dermatitis, but autistic behaviors did not improve [30]. Research on the effects of Mg in ASD has primarily been studied as the effects of its combination with vitamin B6, and as with almost all other supplementation studies, results have been inconsistent [31,32,33,34]. However, more studies are needed to determine the role of the combination of Mg and biotin in the pathophysiology and treatment of ASD.

Propionic acid (PPA), a short-chain fatty acid, is a metabolic fermentation product of some gut bacteria and is often used as an anti-mold agent in the food industry [35,36]. The excessive presence of propionic acid in the blood, cerebrospinal fluid, and neuronal cells is called propionic acidemia, a neurodevelopmental metabolic disorder [37]. High levels of PPA in the blood cross the blood-brain barrier and increase intracellular acidification in neurons. This causes changes in neurotransmitter levels, which are involved in the etiology of neurodevelopmental disorders [38]. In addition, PPA can also cause neurological disorders by affecting synaptic transmission and other forms of neuronal activity [39]. Earlier studies have shown that intraventricular infusions of PPA cause behavioral and brain abnormalities in rats similar to those seen in humans with autism, possibly by altering fatty acid metabolism in the brain [40,41,42]. Investigation of brain tissue from PPA-administrated rats yielded similar findings to what is seen in ASD patients, including reactive astrogliosis, oxidative stress, glutathione depletion, mitochondrial dysfunction, and changes in phospholipid/acylcarnitine profiles [43]. In addition, Choi et al. [6] suggested that in rats treated with 500 mg/kg propionic acid (PPA), PPA administration caused abnormal neural cell organization, which may have led to autism-like behaviors, such as increased aggressive behavior and reduced exploratory behavior activity, and isolative and passive behaviors. Therefore, propionic acid can be used to obtain an autism-like rat model in ASD related animal researches.

As mentioned above, both Mg and biotin are involved in inflammatory and cellular processes. Since these processes have a role in the etiology of ASD, we think that a new molecule, the Mg biotinate (MgB) complex, may provide more improvement in ASD symptoms than Mg and biotin supplementation alone. In addition, it has been determined that MgB is a unique biotin complex with superior absorption and tissue uptake, anti-inflammatory influences, and carboxylase activity compared to D-Biotin [44]. However, no studies on the role of MgB against PPA-induced neurotoxicity have been reported. With the hypothesis that the antioxidant potential of MgB may prevent the adverse effects of PPA on brain function, this study was undertaken to explore the therapeutic effects of MgB on autistic-like behaviors, learning and memory, and possible underlying molecular mechanisms in a propionic acid (PPA)-induced rat model of autism. This study aimed to investigate whether a new molecule, the MgB complex, affects ASD symptoms since both Mg and biotin play a role in inflammatory and cellular processes.

## 2. Materials and Methods

### 2.1. Rats, Study Protocol

In this case, 35 male Wistar rats, 3 weeks old and 80–100 g, were obtained from Fırat University Animal Experiments Research Center (Elazig, Turkey). They were housed under standard conditions (12:12-h light-dark cycle; relative humidity: 50% and 22 ± 2 °C) and had free access to feed and water. The experimental processes were authorized by the Animal Ethics Committee of the Firat University, Elazig, Turkey (2018/15-153) and performed according to the National Institutes of Health’s Guidelines for the Care and Use of Laboratory Animals. 

Animals were randomly distributed into five groups (seven rats per group): (i) Control, rats were treated with 0.9% saline (ii) PPA, rats were administered subcutaneously with PPA (500 mg/kg/day) for 5 days to induce autistic features; (iii) PPA+MgBI, rats were treated with magnesium biotinate (MgBI) (160.7 µg/day) for 2 weeks followed by PPA injection; (iv) PPA+MgBII, rats were treated with MgB (1606.9 µg/day) for 2 weeks followed by PPA injection (v) PPA+MgBIII, rats were given MgB (8034.4 µg/day) for 2 weeks followed by PPA injection. The doses of MgB were calculated based on 10, 100, and 500 mg biotin used for a 70-kg adult human after adjusting doses based on metabolic body size [45,46]. MgB complex contains 90.918% biotin and 9.082 elemental Mg. Therefore, rats were received 160.7, 1606.9, or 8034.4 µg/day of MgB for two weeks orally following PPA treatment. The MgB complex was dissolved in drinking water and offered to rats by oral gavage for 2 weeks, and the complex was provided by Nutrition 21, Inc. (Purchase, NY, USA). The diet used in the study was adapted to contain spray-dried egg whites as the sole protein source. Avidin protein in egg white binds; 1.44 mg biotin/kg purified diet, inhibiting biotin absorption [47]. 

Sodium propionate (PPA) was dissolved in 0.1 MPBS and injected s.c. at a dose of 500 mg/kg, once a day for five days. The PPA does were chosen based on previous studies [6]. Choi et al. [6] found changes in abnormal behavior, histological changes, and gene expression in rats by subcutaneous administration of 500 mg/kg (250 mg/mL, 0.26 M, pH 7.4) daily for 5 consecutive days. They also stated that this method could be used to create an ASD animal model. Rats in the control group were injected with saline. 

### 2.2. Behavioral Tests and Social Interaction Test

Neuro-behavioral tests (Social interaction test, Morris water maze) were performed on day 7 for two consecutive days. The neurological, sensory, and behavioral scoring was carried out by a blinded evaluator (the time spent in non-social and social interaction in reciprocal social interaction test). 

As previously described, a three-chamber social interaction test was utilized to determine the sociability and social novelty of animals from all groups [6]. Briefly, the rats were habituated to the environment for 5 min before the test began, and then each of them was placed in the middle of the compartment. Age and sex-matched a stranger rat (habituated to the wire cage in the apparatus) was placed in a small wire cage in either left or right compartment for the sociability phase, and this compartment was designated as a stranger zone. The other zone was left empty, and this compartment was chosen as an empty zone. The animals were observed for a 10-min sociability test. The time spent in stranger zone 1 and empty zone were recorded. Then the ratio of the time spent in stranger zone 1 and the time spent in the empty zone was calculated (Sociability Index). The social preference test was carried out after the sociability test. Another strange rat was placed into the empty zone, and this zone was described as stranger zone 2. The social preference index, which is the ratio of the time spent in stranger zone 1 and stranger zone 2, was calculated. The device was cleaned with 50% ethanol between test periods [6].

### 2.3. Morris Water Maze Task

The Morris water maze (MWM) test assessed spatial learning and memory. The MWM test was carried out in a dimly lit room using a 120 cm diameter circular black tank at 50 cm depth. The tank was filled with water to a height of 40 cm, and the temperature of the water was adjusted to 24 ± 2 °C. A 10 cm diameter platform was placed in the center of a quadrant of the tank and submerged 1.5 cm below the water surface. The platform was initially concealed in a randomly selected section within the tank, but care was taken to be in the same place during the 5-day test. Four starting positions (north, south, east, west) were spaced around its perimeter, dividing the pool into four equal quadrants. Each rat was allowed to find the submerged platform within 60 s and rest on it for 30 s. If the rat failed to find a hidden platform within the assigned time, it was placed on the platform for 10 s. The rat was then returned to a heated cage for 5 min between trial intervals, and then the rat would be placed in water at a different start location to repeat the same process. At the end of the fifth day, a probe test was carried out. For this purpose, the platform was removed from the water, and the rats were left into the middle of the tank to record the movements for 60 s. Where the platform was earlier located, the time spent by the rat was defined.

### 2.4. Laboratory Analyses

At the end of the study, the rats were sacrificed by cervical dislocation under anesthesia. The blood and brain samples were taken and stored at −80 °C for further analyses.

To determine serum and brain Mg levels, nearly 0.3 g brain and 0.5 mL serum were digested with 5 mL concentrated nitric acid in a Microwave Digestion System (Berghof, Eningen, Germany) for 30 min, and diluted 1∶10 with distilled deionized water. Mg concentrations were assessed by flame AAS (AAS, Perkin-Elmer, Analyst 800, Norwalk, CT, USA) via identified and thoroughly verified processes at the 285.2 nm wavelength in the samples. The method was verified with certified reference materials, and the accuracy was 2%. 

The serum and brain biotin concentrations were measured by HPLC (Shimadzu, Kyoto, Japan) as previously described with minor modifications [47]. The reversed-phase column used was a C18-ODS-3 column (250 × 4.6 mm, 5 m), and the biotin-containing chromatography fractions were dried under a stream of nitrogen before the test. 

For the determination of serotonin, dopamine and oxidative stress-related biochemical parameters, malondialdehyde (MDA), superoxide dismutase (SOD), catalase (CAT) and glutathione peroxidase (GPx), brain samples were weighed and homogenized in phosphate buffer to give 20% (*w/v*) homogenate, and the supernatant obtained by centrifugation (3000× *g* for 10 min). Serotonin and dopamine levels were analyzed by ELISA using a commercially available assay kit (Immuno-Biological Laboratories, Hamburg, Germany). Serum and brain MDA was measured as defined by Olszewska-Słonina et al. [48]. GPx was determined as explained by Giustarini et al. [49]. SOD and CAT activities were estimated as defined by Serra et al. [50,51] and Hadwin [50,51], respectively.

### 2.5. Western Blot Analyses

Brain samples were homogenized and sonicated in lysis buffer. Total protein was measured by a NanoDrop apparatus (MaestroGen, Las Vegas, NV, USA). 20 μg protein were size-fractionated using any-kD Mini-Protean TGX gel electrophoresis and then transferred to a nitrocellulose membrane, and membranes were blocked and washed in TBS-T, and incubated overnight with AC, ACC1, ACC2, PC, PCC, MCC, presynaptic synapsin I, postsynaptic PSD95, PSD93, IL-17, IL-6, TNF-α, NFkB, GFAP, GAP43, ICAM-1, BDNF, CXCL 16, OPG and MMP-9 antibodies (Abcam, Cambridge, UK). The following day, membranes were washed and then incubated with horseradish peroxidase-conjugated goat anti-rabbit (Abcam, Cambridge, UK) or goat anti-mouse (Abcam, Cambridge, UK) secondary antibody (diluted 1:1000) for 1 h at room temperature. Protein loading was controlled with an anti-β-actin antibody (Abcam, Cambridge, UK). Protein levels were analyzed densitometrically using an image analysis system (Image J; National Institute of Health, Bethesda, MD, USA), corrected with values determined on β-actin blots, and expressed as relative values compared with the control group.

### 2.6. Histological Evaluation

Brain samples were removed and post-fixed in 4% paraformaldehyde solution and stored at 4 °C for 24 h, succeeded by 70% isopropanol overnight. Then, brains were placed in an 18% sucrose solution for cryoprotection prior to sectioning or paraffin embedding. Haematoxylin and eosin (H&E) staining were used to evaluate the pathological changes. 

### 2.7. Statistical Analyses

Data were expressed as mean ± SEM. The sample size was evaluated based on a power of 85% and a *p*-value of 0.05. A sample size of seven per treatment was calculated. The data were analyzed using the GLM procedure of SAS (SAS Institute: SAS User’s Guide: Statistics). The treatments were compared using ANOVA and Student’s unpaired *t*-test; *p* < 0.05 was considered statistically significant.

## 3. Results

### 3.1. Effects of MgB Supplementation on Mg, Biotin, Serotonin and Dopamine Concentrations

As shown in Table 1, PPA treatment reduced (*p* < 0.05) biotin and Mg levels (in the serum and brain), as well as brain serotonin and dopamine concentrations compared with control. Of great interest, all these parameters were significantly improved after MgB administration. In groups treated with MgB, the highest dose, Mg, serotonin values were found in the PPA+MgBIII group, and the highest dopamine level was found in the PPA+MgBI group (Table 1).

### 3.2. Effects of MgB Supplementation on Oxidative Stress-Related Biochemical Parameters

PPA-induced oxidative stress is characterized by a significant increase in MDA level (in the serum and brain) and low brain CAT, GPx, GSH, and SOD levels compared to the control group (Table 2). Remarkably, MgB supplementation significantly reduced MDA concentration in a dose-dependent manner. In addition, the activities of antioxidant enzymes (CAT, GPx, SOD, and GSH) were significantly improved (*p* < 0.05) by MgB supplementation, the highest dose being the most effective (Table 2).

### 3.3. Effects of MgB Supplementation on Brain Histology

Firstly, brain tissues were examined macroscopically, and no difference was found among groups (Figure 1A). Next, histological examination in the hippocampus of PPA-treated rats exhibited disorganization of the external granular and pyramidal layers and a reduced granule cell layer compared with control rats. However, MgB treatment ameliorated this reduction and disorganization, with a remarkable effect observed in rats administered with the highest dose of MgB (Figure 1B). Reduction in the number of Purkinje cells, nucleus density, edema of Purkinje layer, and cytoplasmic edema of Purkinje layer in the cerebellum were observed in PPA-treated rats, compared with control. MgB supplementation ameliorated these alterations in the PPA-induced autism rats (Figure 1C). Nissl staining of hippocampal CA1 region results showed that the number of Nissl-positive cells was reduced in PPA-induced autism rats compared to the control group (Figure 1D). 

### 3.4. Effects of MgB Supplementation on Social Behavior, Learning, and Memory

Analyzes of social behaviors, learning, and memory showed that center duration (Figure 2A), sociability index (Figure 2B), social preference indices (Figure 2C) were significantly decreased in the PPA group as compared with the control group, which indicated anxiogenic behavior (*p* < 0.001). In addition, the PPA-treated rats without MgB administration had a decrease in entries to the target quadrant (*p* < 0.001; Figure 2D) and probe trial (*p* < 0.001; Figure 2E) compared to the control, indicating that they had poorer learning ability. Treatment with MgB reduced anxiogenic-like behavior (*p* < 0.05), which, the highest dose of MgB showed more efficacy than lower doses of MgB as compared to the PPA group (*p* < 0.05). We also noticed that in all groups (except the PPA+MgBI group, from day 4 to day 5), the latency decreased gradually from the first day to the fifth day. Similar to other parameters, MgB was acted dose-dependent (Figure 2F). The highest dose of MgB was more effective. These results suggest that MgB treatment may ameliorate spatial learning and memory function in the PPA-induced autism-like rats.

### 3.5. Effects of MgB Supplementation on Neurotoxicity-Related Cytokines

As shown in Figure 3, PPA treatment significantly up-regulated TNF-α, IL-6, IL-17, CCL-3, CCL-5, and CXCL-16 in the brain but decreased the expression levels of OPG and MMP-9, compared to the control group (*p* < 0.05). However, MgB supplementation prevented neurotoxicity by significantly reducing (in a dose-dependent manner) the expression level of TNF-α, IL-6, IL-17, CCL-3, CCL-5, and CXCL-16 in the brain, compared to the control group. The expression levels of OPG and MMP-9 were significantly increased after MgB treatment (Figure 3).

### 3.6. Effects of MgB Supplementation on Neurodevelopment Markers

Compared to the control group, PPA induced a significant decrease in brain ACC-1, ACC-2, PC, PCC, and MCC expressions. The expression levels of these markers were significantly increased in the brain of PPA-induced autism-like rats treated with MgB. MgB acted in a dose-dependent manner (Figure 4).

### 3.7. Effects of MgB Supplementation on Neurotransmission-Related Markers

PPA treatment significantly decreased the expression levels of BDNF, GAP, ICAM, PSD-93, and PSD-95 in the brain but up-regulated the brain GFAP, compared to the control group. PPA-induced autism-like rats administered with MgB (at all doses) exhibited a significant increase in the expression levels of BDNF, GAP, ICAM, PSD-93, and PSD-95 in the brain and decreased GFAP expression, compared to the control group. The neuroprotective effect of MgB was dose-dependent (Figure 5).

## 4. Discussion

The present study showed that MgB supplementation improved histological and biochemical changes as well as cognition and autistic-like behaviors in an animal model of autism spectrum like-disorders based on exposure to PPA. This improvement was characterized by the significant increase in the brain and serum Mg, biotin, serotonin, dopamine concentrations, and antioxidant enzymes. Control rats spent more time in the center chambers during the sociability test, reflecting normal sociability. However, compared to control rats, which showed less sociability in PPA-treated animals, PPA rats reduced time spent in the stranger chamber and increased time spent in the empty chamber. Administration of MgB has significantly attenuated PPA-induced reduction in time spent in stranger chambers. Results on the effect of PPA on social behaviors were similar to previous studies [6,52]. However, there was no literature on how MgB affected these animals in PPA-induced rats with autism spectrum like-disorders. In addition, MgB treatment significantly improved learning and memory and regulated the neurotoxicity-related cytokines and neurotransmission-related markers. 

Although the prevalence of ASD is increasing day by day, the data about its etiology and treatment is limited. Various technics/chemicals such as folic acid, thalidomide, valproic acid, propionic acid, and D2 dopamine receptor heterozygous knockout model are used to experimentally induce autistic features in rodents [53,54,55,56,57,58]. Numerous studies have proven that the VPA model can be used as an essential tool to examine the underlying mechanism of ASD [59]. However, this model also has some limitations [60]. For instance, most patients with ASD have not been exposed to this drug, and previous research has determined that VPA exposure in early pregnancy can cause nerve cell defects and congenital malformations rather than ASD [60]. By penetrating physiological lipid barriers such as blood-brain and intestinal-circulatory barriers, PPA can cause various central nervous system disorders by increasing the release of neurotransmitters and intracellular acidosis [61]. A previous study [6] stated that PPA (500 mg/kg, s.c) was responsible for disrupting neural cell organization. In the present study, we used propionic acid to easily induce Autism Spectrum Disorder in rodents similar to human autism [57]. 

As reported previously, we noticed in the current study that PPA treatment easily induced autistic features in rats after 21 days of exposure [6,57,62]. PPA causes severe damage to brain structure and alters dopamine, serotonin, and glutamate pathway mainly through intracellular calcium overproduction [63,64]. In the current study, PPA treatment significantly reduced the number of Purkinje cells, nucleus density, and increased edema of Purkinje layer as well as cytoplasmic edema of Purkinje layer. Prior studies have stated that cerebellar abnormalities are related to non-motor diseases such as OSB and motor diseases [65,66]. Similar to the results of the current study, in the cerebellum of patients with ASD, reduced volume and loss of pyramidal neurons and granule cells have also been reported [4]. However, it is still unclear how the cerebellum might contribute to core symptoms of ASD, such as abnormalities in social interactions and repetitive behaviors. In addition, there were histopathological changes in the hippocampus of 3-week-old PPA-induced autism-like rats. It is identified that the hippocampus, which is impaired in ASD, has key role in social interaction, memory, and spatial reasoning [67]. Moreover, the 18-24 month period, when ASD symptoms begin to emerge, is a vital milestone in hippocampal development. Given these findings, Banker et al. [67], have postulated that deficiency in changed hippocampal structure and function may affect occurring deficits in social interaction, memory, and spatial reasoning in ASD. In some studies, it was reported that hippocampal volume increased in children with ASD compared to typically developed children, however in some studies, there was decreased volume or no significant difference in hippocampal volume in children with ASD compared to typically developed children [68,69,70,71,72]. Uppal et al. [73] showed increased perforated synapses in the CA1 of the 5-week-old Shank3 mouse model of autism but found that this did not persist with age. In another study with a similar rat model, it was found that rats had low spine density in CA1 at 4 weeks of age but were comparable to the control group at 10 weeks of age [74]. These changes, which may occur in some patients with autism during the maturation of the hippocampus in the early stages of life, may affect the social interaction deficits in these patients. In addition, these structural alterations observed in the brain of autistic patients may negatively affect the synthesis and function of neurotransmitters involved in ASD, such as serotonin and dopamine. However, these structural changes observed in the brains of patients with autism were not a consistent feature, and no discernible relationship was found between these structural changes and the clinical features/severity of autism.

Neurotransmitters and neuropeptides are involved in normal brain development, neuronal cell migration, differentiation, synaptogenesis, apoptosis, and synaptic pruning. Therefore, it is thought that neurotransmitters and neuropeptides may play a role in the etiology of ASD [75]. Studies have shown that autistic children and animal models have higher levels of the serotonin transporter (SERT or 5-HTT), or serotonin, compared to controls [76,77,78]. In a meta-analysis study conducted by Gabriele et al., it was stated that elevated 5-HT blood levels might be one of the potential marker candidates [79]. The current study found low serotonin levels in the brain of PPA-induced autism-like rats compared to the control animals. These results disagree with a previous work that showed that rats administered with thalidomide and valproic acid (teratogen agents used to induce autism in rats) exhibited high serotonin and dopamine levels in the brain compared to the control rats [80]. This difference may be due to the type of teratogen agents used and age differences. Indeed, in the work of Narita et al. [80], thalidomide and valproic acid were administered during embryonic days (days 2–11), while in the current study, PPA was given to 3-week-hold pups for five consecutive days. Consistent with the present study, a low serotonin transporter level and decreased serotonin and dopamine levels in the brain of patients with autism have been reported [81,82]. However, MgB supplementation induced a significant increase in serotonin levels compared with untreated PPA-induced autism-like rats. The better learning and memory observed in those receiving higher MgB therapy may be due to increased serotonin levels. Since antipsychotic drugs are effective on some symptoms of autism, the researchers suggest that dopaminergic dysfunctions might be associated with ASD [83]. However, data supporting the role of dopaminergic dysfunction in the etiology of ASD are limited. Dichter et al. [84] demonstrated that ASD is correlated with dopamine imbalances in specific brain religions. However, the homovanillic acid (HVA) measurements, a dopamine metabolite, were inconsistent [85]. The studies have shown that HVA levels may be low, normal, or high in ASD [85]. In the current work, the high dopamine level observed in the brain of PPA-induced autism-like rats was significantly lowered after MgB treatment, indicating a normal dopamine release in the cortex and an improvement in neuronal response in the nucleus accumbens.

The improvement in serotonin and dopamine levels in the brain and serum of MgB-treated rats was correlated with increased Mg and biotin contents (in the brain and serum). The role of biotin and Mg in the inflammation process has been shown [86,87]. It has been found that in biotin-deficient conditions, immune system cells decreased while increased levels of proinflammatory cytokines are observed [88]. Similarly, leukocyte and macrophage activation, the release of inflammatory cytokines and acute-phase proteins as well as excessive production of free radicals have been detected in Mg deficiency conditions [86]. Thus, the ability of MgB to increase Mg and biotin levels in the brain and serum of PPA-induced autism-like rats may suggest a beneficial effect on neuroinflammation and oxidative stress, which are the critical processes involved in ASD. For this reason, it was of great interest to investigate the impact of MgB on oxidative stress enzymes and neuroinflammation markers.

The imbalance between oxidants and antioxidants is another factor that has been reported to affect the emergence of ASD. This imbalance causes the overproduction of reactive oxygen species (ROS), leading to oxidative stress. Pieces of evidence suggest that ROS may have a possible role in the pathophysiology of autism [89]. In experimental animal studies, PPA has been shown to decrease antioxidant enzyme levels and increase oxidative products, which results in autism-like findings [8]. Similarly, in the current study, it was found that PPA decreased CAT, SOD, GPx, GSH activities and increased lipid peroxidation (high MDA level). In parallel, MacFabe et al. [8] have reported that the intravenous administration of PPA elevated lipid peroxidation and decreased antioxidant enzymes levels, associated with cognitive disability and memory alterations in rats. In this study, the emergence of autism-like features in rats may be related to this increased oxidative stress. For instance, we found that PPA treatment negatively affected the social behavior and Morris water maze task parameters by significantly decreasing the center duration, sociability index, social preference indices, as well as entries to target quadrant and probe trial. Of great interest, MgB significantly reduced oxidative stress in PPA-induced autism-like rats, characterized by the significant increase in antioxidant enzymes (CAT, SOD, GPx, GSH) and a decrease in lipid peroxidation (low MDA level). Moreover, MgB enhanced learning and memory in PPA-induced autism-like rats in a dose-dependent manner. In a similar study, Gao et al. [90] have reported that docosahexaenoic acid (150 and 300 mg/kg) treatment improved learning and memory (dose-dependent manner) of valproic acid-induced autistic-like rats.

Neuroinflammation-related markers such as TNF-α, interleukins (IL-6, IL-17), and chemokines (CCL-3, CCL-5, and CXCL-16) have been involved in the etiology of ASD [11,12,13]. Wei et al. [14] determined that overexpression of IL-6 in the brain of the mouse model developed severely impaired dendritic spine development and contributed to the development of autistic-like features. IL-6 signaling is also involved in the maternal immune system activation associated with autism [91]. Chemokines (CCL-3, CCL-5, CXCL-16) regulate CNS inflammatory states or neurogenesis [15]. The current study demonstrated that PPA-induced autism in rats was characterized by a significant increase in the expression of TNF-α, IL-6, IL-17, CCL-3, CCL-5, and CXCL-16 levels as a down-regulation of OPG in the brain of PPA-induced autism-like rats, compared to the control animals. It was observed that rats with more proinflammatory cytokine production had more oxidative stress and impaired cognitive function in their brains. This may be due to the neurotoxicity of PPA, as previously reported [7]. These results are consistent with the work of MacFabe et al. [8], who noted that PPA-treated animals exhibited high proinflammatory cytokines (IL-6 and TNF-α) after treatment. Moreover, the overexpression of interleukins (IL-6, IL-8, and IL-12p40) and chemokines (CCL2, CCL5, and CCL11), as well as autistic-like phenotypes (reduced social interactions and impaired juvenile play), was also found in autistic mice [92,93]. Remarkably, the expression of proinflammatory cytokines and chemokines in the brain of PPA-induced autism-like rats was down-regulated after MgB treatment, probably due to its antioxidant potential. Since OPG (a member of the TNF receptor subfamily) inhibits osteoclastogenesis by binding the receptor activator of NF-κB ligand (RANKL) to inhibit RANKL interaction with its receptor RANK, the up-regulation of OPG in the brain of MgB-treated rats may imply that MgB modulated pro-inflammatory cytokines through OPG–RANKL–RANK pathway [94].

In the present study, PPA administration decreased the brain levels of the biotin-dependent carboxylases such as ACC1, ACC2, PC, PCC, and MCC. These carboxylases participate in many metabolic pathways such as fatty acid synthesis, amino acid, cholesterol, and odd-chain fatty acids metabolism, gluconeogenesis, and tricarboxylic acid anaplerosis [95]. However, MgB administration increased brain ACC1, ACC2, PC, PC, PCC, and MCC levels in PPA-induced autism-like rats. To our knowledge, the present results are among the first to demonstrate the effects of MgB in stimulating biotin-dependent carboxylases in the brain of PPA-induced rats.

Synaptic plasticity is vital for hippocampal-dependent learning and memory and can be measured using long-term potentiation or synaptic morphology. MMP-9, BDNF, GAP 43, ICAM, Synapsin-I, PSD-93, and PSD-95 are essential markers in neural system development and neuroplasticity [17,18]. MMP-9 establishes synaptic connections during the development and restructuring of synaptic networks in the adult brain [96]. BDNF inhibits apoptosis and promotes neuronal reorganization, while ICAM improves tissue formation, maintenance, and function [19]. Synapsin I is a vesicle-associated protein involved in the regulation of synaptogenesis. Mutations in synapsin I was detected in patients with ASD [97]. PSD-95 and PSD-93 are membrane-associated guanylate kinases that form NMDA receptor-associated signaling complexes involved in synaptic plasticity [98]. GFAP is involved in the astroglial cell activation observed in central nervous system injury and neurodegeneration conditions. Thus, it is stated that GFAP may be used as a marker for brain damage as observed in ASD patients [99,100]. Our study showed that PPA treatment significantly decreased the expression of MMP-9, BDNF, ICAM, Synapsin I, PSD-93, and PSD-95 but elevated GFAP in the brain of PPA-induced PPA-induced autism-like rats. The down-regulation of MMP-9 in the brain of PPA-induced autism-like rats is unexpected because elevated brain MMP-9 levels after PPA exposure amplify neuroinflammation by activating TNF-α and other pro-inflammatory markers and promote autistic-like features [13]. The high level of GFAP in the brain of autistic animals is expected because increased GFAP in the cerebellum has been reported in an individual with autism [101]. Interestingly, the neuroplasticity-related markers were significantly improved after MgB treatment, which implies that MgB protects the brain against the cytotoxic effect of PPA. The upregulation of neuroplasticity-related markers observed in the brain of MgB-treated rats could indicate a hyperplasticity state within the CNS that eventually leads to an improved clinical phenotype. In parallel, Slutsky et al. [102] have reported that Mg enhances synaptic plasticity in cultured hippocampal neurons, suggesting its role as a positive regulator of synaptic plasticity. In addition, Xu et al. [103] showed that Mg sulfate up-regulates synapsin I, PSD95, and PSD93 in the brain and improves learning and memory in streptozotocin-induced sporadic Alzheimer’s model rats.

## 5. Conclusions

This study demonstrates that MgB improves autistic-like features in PPA-induced autism-like rats by decreasing oxidative stress and pro-inflammation cytokines and enhancing learning and memory. MgB can be considered a pharmacotherapeutic agent that may positively affect biochemical, behavioral, and molecular changes associated with autism. However, it should be noted that this study was performed in PPA-induced autism-like rats, and further studies are needed to determine how MgB will affect patients with autism.

## Figures and Tables

**Figure 1 nutrients-14-01280-f001:**
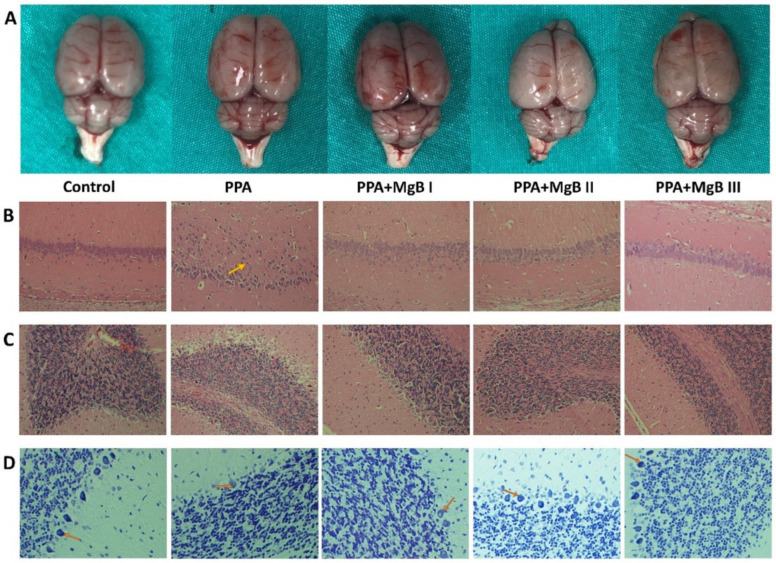
(**A**) Effects of MgB supplementation on macroscopy of the brain in PPA-induced autism in rats. (**B**) Effects of MgB supplementation on the histopathology of the hippocampus CA1 fields (H&E 200×) in PPA-induced autism in rats. There was granule cell layer thickness in the hippocampus of PPA-exposed rats. Treatment with MgB at different doses ameliorated this reduction in the PPA-exposed rats. (**C**) Effects of MgB supplementation on the histopathology of the cerebellum (H&E 200×) in PPA-induced autism in rats. Reduction in the number of Purkinje cells, nucleus density, edema of Purkinje layer, and cytoplasmic edema of Purkinje layer was observed in PPA- exposed rats. Treatment with MgB ameliorated these alterations in the PPA-exposed rats. (**D**) Effects of MgB supplementation on Nissl staining of the hippocampal CA1 fields (400×) in PPA-induced autism in rats. The number of Nissl-positive cells was reduced in the hippocampal CA1 region of PPA-exposed rats compared to the control group. Treatment with MgB ameliorated these changes in the PPA-exposed rats.

**Figure 2 nutrients-14-01280-f002:**
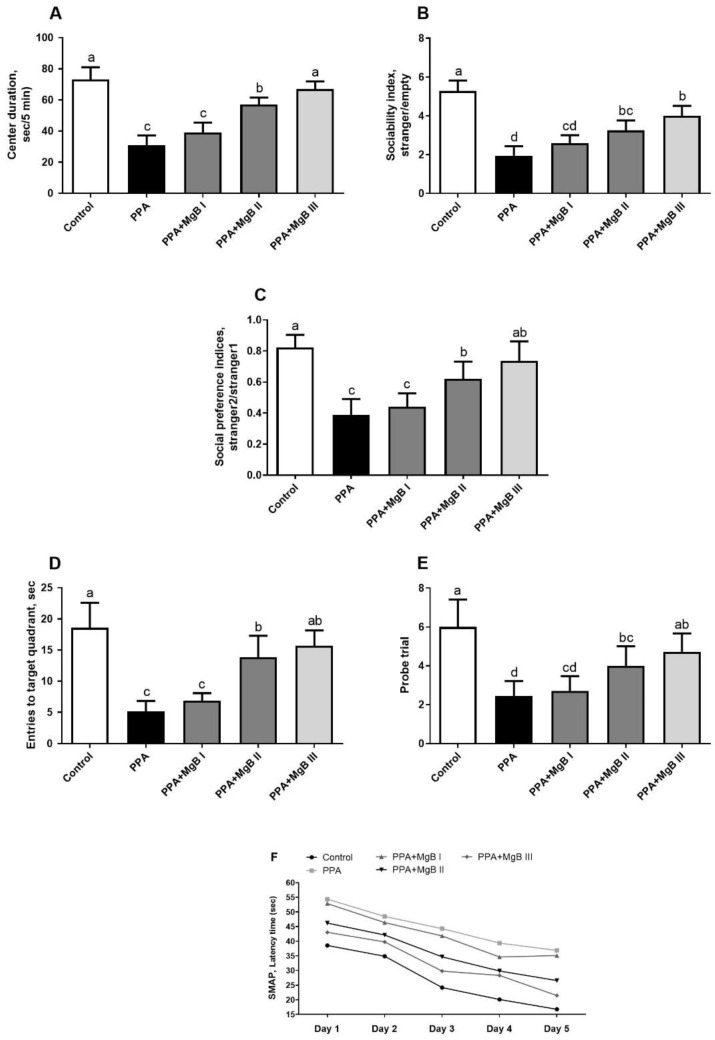
Effects of MgB supplementation on social behavior tests (Center duration (**A**), Sociability index (**B**) and Social preference indices (**C**)) and Morris water maze task (entries to target quadrant (**D**), probe trial (**E**), and latency to find the hidden platform from the first day to the fifth day (**F**)) in the in PPA-induced autism in rats. a–d: Values within the bars with different superscripts are significantly different (Tukey’s post-hoc test, *p* < 0.05).

**Figure 3 nutrients-14-01280-f003:**
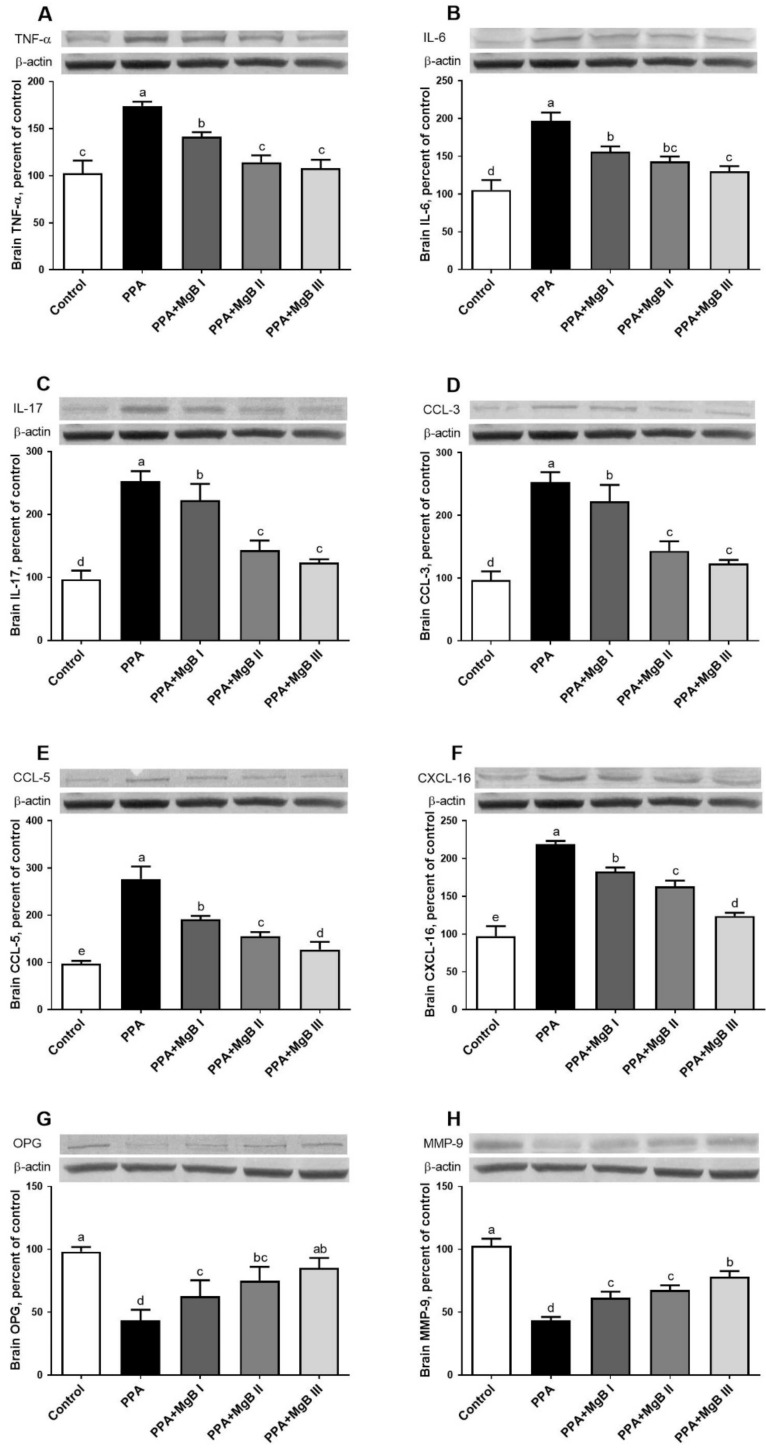
Effects of MgB supplementation on brain tumor necrosis factor-alpha (TNF-α, (**A**)), interleukin 6 (IL-6, (**B**)), interleukin 17 (IL-17, (**C**)), chemokine (C-C motif) ligand 3 (CCL-3, (**D**)), chemokine (C-C motif) ligand 5 (CCL-5, (**E**)), chemokine (C-X-C motif) ligand 16 (CXCL-16, (**F**)), brain osteoprotegerin (OPG, (**G**)) and matrix metallopeptidase 9 (MMP-9, (**H**)) protein levels in PPA-induced autism in rats. Data are shown as a percent of the control value. Blots were repeated at least three times. a–e: Values within the bars with different superscripts are significantly different (Tukey’s post-hoc test, *p* < 0.05).

**Figure 4 nutrients-14-01280-f004:**
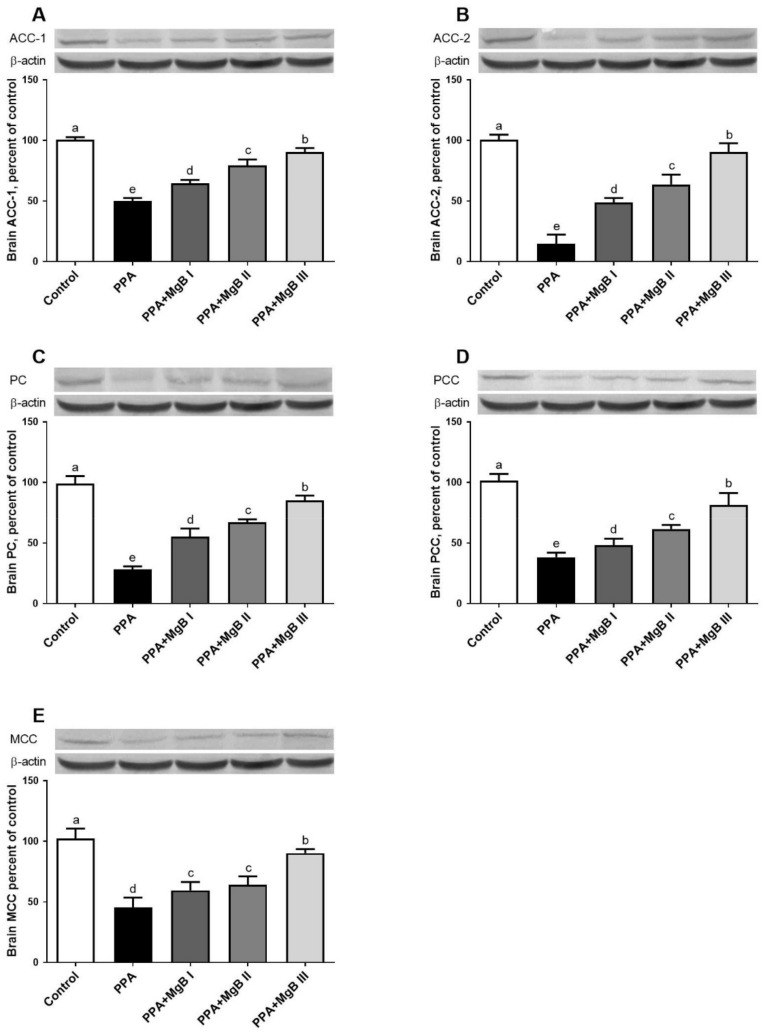
Effects of MgB supplementation on brain acetyl CoA carboxylase 1 (ACC-1, (**A**)), acetyl CoA carboxylase 2 (ACC-2, (**B**)), pyruvate carboxylase (PC, (**C**)), propionyl-CoA carboxylase (PCC, (**D**)), and 3-methylcrotonyl-CoA carboxylase (MCC, (**E**)) protein levels in PPA-induced autism in rats. Data are expressed as a percent of the control value. Blots were repeated at least 3 times. a–e: Values within the bars with different superscripts are significantly different (Tukey’s post-hoc test, *p* < 0.05).

**Figure 5 nutrients-14-01280-f005:**
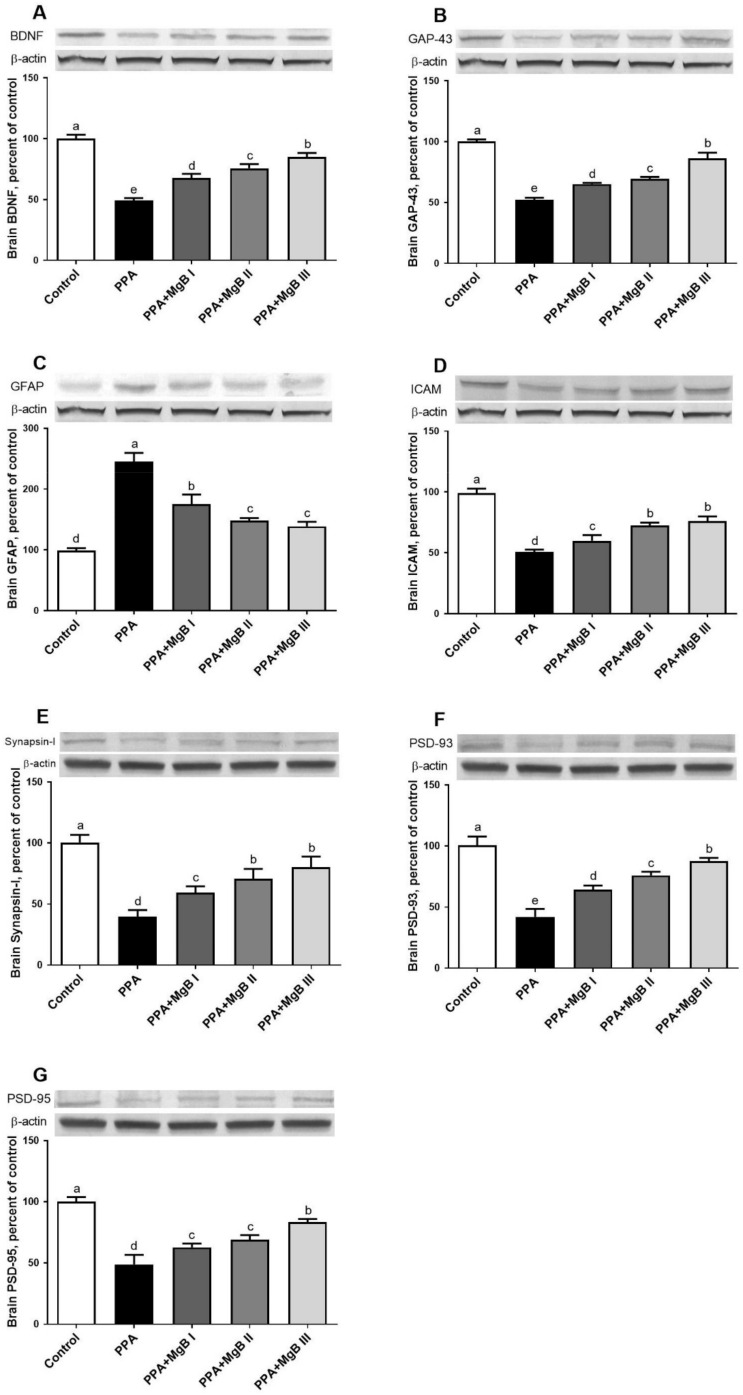
Effects of MgB supplementation on the brain-derived neurotrophic factor (BDNF, (**A**)), growth-associated protein (GAP-43, (**B**)), glial fibrillary acidic protein (GFAP, (**C**)), intercellular adhesion molecule 1 (ICAM, (**D**)), Synapsin I (**E**), postsynaptic density protein 93 (PSD-93, (**F**)) and postsynaptic density protein 95 (PSD-95, (**G**)) protein levels in PPA-induced autism in rats. Blots were repeated at least 3 times. a–e: Values within the bars with different superscripts are significantly different (Tukey’s post-hoc test, *p* < 0.05).

**Table 1 nutrients-14-01280-t001:** Effects of magnesium biotinate (MgB) supplementation on magnesium (mg/dL or µg/g), biotin (nmol/L or nmol/g), brain serotonin (ng/g), and dopamine (ng/g) concentrations in propionic acid (PPA) induced autism in rats.

Items	Groups				
	Control	PPA	PPA+MgB I	PPA+MgB II	PPA+MgB III
Serum Mg	3.36 ± 0.14 ^c^	2.20 ± 0.08 ^e^	2.76 ± 0.05 ^d^	4.33 ± 0.06 ^b^	5.88 ± 0.12 ^a^
Brain Mg	143.84 ± 1.97 ^b^	97.12 ± 2.99 ^d^	113.24 ± 5.11 ^c^	144.21 ± 3.58 ^b^	168.13 ± 4.28 ^a^
Serum Biotin	72.30 ± 1.13 ^c^	49.16 ± 1.80 ^d^	55.81 ± 3.40 ^d^	99.94 ± 2.43 ^b^	121.50 ± 3.06 ^a^
Brain Biotin	0.159 ± 0.006 ^bc^	0.099 ± 0.007 ^d^	0.146 ± 0.007 ^c^	0.174 ± 0.006 ^ab^	0.187 ± 0.007 ^a^
Serotonin	54.85 ± 0.94 ^a^	11.24 ± 0.47 ^e^	17.07 ± 0.90 ^d^	27.20 ± 1.24 ^c^	36.95 ± 0.59 ^b^
Dopamine	8.44 ± 0.88 ^d^	36.61 ± 2.01 ^a^	29.03 ± 1.26 ^b^	18.58 ± 1.04 ^c^	12.64 ± 0.70 ^d^

PPA: Propionic acid, MgB: Magnesium biotinate. Data presented as mean and standard error. Different superscripts (a–e) indicate group mean differences (*p* < 0.05).

**Table 2 nutrients-14-01280-t002:** Effects of magnesium biotinate (MgB) supplementation on serum and brain malondialdehyde (MDA) levels (nmol/mL or nmol/mg), brain antioxidant enzymes activities (U/mg protein), and glutathione (GSH, µ/g protein) in propionic acid (PPA) induced autism in rats.

Items	Groups				
	Control	PPA	PPA+MgB I	PPA+MgB II	PPA+MgB III
Serum MDA	0.57 ± 0.04 ^e^	1.93 ± 0.03 ^a^	1.65 ± 0.04 ^b^	1.41 ± 0.02 ^c^	0.93 ± 0.03^d^
Brain MDA	1.84 ± 0.03 ^e^	5.96 ± 0.17 ^a^	4.60 ± 0.12 ^b^	3.80 ± 0.04 ^c^	2.86 ± 0.08 ^d^
CAT	32.53 ± 1.69 ^a^	10.12 ± 0.46 ^d^	13.42 ± 0.87 ^d^	20.77 ± 0.64 ^c^	26.56 ± 0.56 ^b^
SOD	145.36 ± 1.96 ^a^	51.17 ± 2.07 ^e^	64.10 ± 2.94 ^d^	93.51 ± 2.38 ^c^	126.47 ± 2.78 ^b^
GPx	26.29 ± 0.88 ^a^	7.71 ± 0.31 ^d^	9.48 ± 0.50 ^d^	14.19 ± 0.79 ^c^	19.82 ± 0.99 ^b^
GSH	68.65 ± 2.22 ^a^	28.03 ± 1.62 ^d^	31.92 ± 1.83 ^d^	42.15 ± 1.92 ^c^	53.19 ± 2.25 ^b^

PPA: Propionic acid, MgB: Magnesium biotinate MDA: Malondialdehyde; CAT: Catalase; SOD: Superoxide dismutase GPx: Glutathione peroxidase; GSH: Glutathione. Data presented as mean and standard error. Different superscripts (a–e) indicate group means differences (*p* < 0.05).

## Data Availability

The data presented in this study are available on request from the corresponding author.

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
