# Peer review of "Therapeutic Effects of a Novel Form of Biotin on Propionic Acid-Induced Autistic Features in Rats"

_nutrients, 2022, doi:10.3390/nu14061280_

Round 1

Reviewer 1 Report

Re: nutrients-1552763

Sahin et al have examined the effects of a novel formulation of biotin on a propionic acid-induced autism model in rats. There are findings of interest, but there are problems with the communication of a number of aspects.  Also, the understanding of the salience to autism needs significant improvement.

First, the Abstract discusses ‘improves autistic behaviors’ but the results in the Abstract only mention ‘improved learning and memory’- need to mention the social results in the Abstract in order to comment on autism-associated behaviors at the end. Also, can’t discuss in terms of ‘managing autism’- these findings have no bearing on the clinical management of autism at this stage.

Introduction, line 7 to 8 change ‘supposed’ to ‘believed’.  Paragraph 4 change ‘The using rates of’ to ‘The rates of use of’.  The authors need to discuss the clinical salience of the PPA model for autism.  How is this recognized as an autism model? Why?  This is not done, leaving the paper only having implications for the PPA toxicity, which also just happens to have a social component.  This must be addressed.  Also, I’m not sure why the authors need to dive into the controversial complementary treatments discussion.  This study involves a very specific treatment targeting one particular mechanism- you don’t need to make controversial statements about whether a broad range of these therapies are ‘acceptable for the management of ASD’ in general, or even the recommendation or not for Mg and B6.  Also, the authors need to discuss more about why they are using Mg biotinate in contrast other formulations such as D-Biotin.  The only thing stated is one brief sentence at the beginning of the final paragraph.

Methods- need to discuss the model to introduce the reader to the appropriateness of this dose of PPA, and why the various doses of MgB were chosen.  Please rephrase ‘as clarified earlier noted’.  Were the test rat and the stranger rat sex matched?  The Morris Water Maze Task description needs clarification- do the authors mean ‘functionally’ instead of ‘fictionally’?  What is meant by ‘were left in the water from each partition wall which was determined as imaginary’ and ‘it was kept on the platform and kept there for 30 seconds’ and ‘Where the platform was earlier located, the time spent by the rat was defined.’?

Results- Another salience aspect- ‘severe neuronal loss and degeneration in the hippocampus’ is not at all what is seen in autism, but is found with this model in section 3.1.  Purkinje cell reduction is seen in ASD, however.  Important to link the findings specifically to ASD, and show an understanding of when it matches and when it does not match if this is to be the focus.  Also- ‘the appearance of the Purkinje layer and cells is maintained following MgB treatment’- the phrasing sounds like the defects were maintained rather than ameliorated – please rephrase.  The text in section 3.1 and Figure 1 states the doses of MgB as 1, 100 and 500 but the rest of the paper says 10, 100, and 500- which is it?  The increased central time in the control condition in Figure 2 seems very odd- might discuss whether there is less movement the PPA group.

Discussion- First paragraph ‘MgB could be considered an effective agent in the management/treatment of autism’- this is an extreme overstatement of what is supported by this data.  The reversal of the effects of a biochemical challenge during pregnancy that might be an ASD rat model cannot lead to clinical conclusions.  In the second and third paragraph, there is not a clear understanding of autism models- should review the mainstream models to understand what the leading models are to discuss at the beginning- and again the salience of the PPA model is not presented.  Change ‘severe damages’ to ‘severe damage’.  Please note that the massive changes in the Purkinje cells and the neuronal loss and degeneration in the hippocampal CA1 region’ do not match well the pathology seen in autism, and there is no such thing as ‘autistic rats’.  The statement ‘The relationship between serotonin and dopamine levels and autism have not been clearly defined’ is misleading- they are rather heavily studied, and the authors should review this evidence, and hyperserotonemia is not considered a primary cause of autism by most investigators.  While it is true that antipsychotics are helpful, the support for major dopaminergic dysfunction as central in ASD is very limited at best. Later ‘One of the proposed hypotheses to explain the etiology of ASD is the imbalance between oxidants and antioxidants’- this maybe be an important factor, at least in a subset of cases, but the way this is stated sounds like it is a primary cause, which is an overstatement.  Not sure how the findings herein are ‘implying vulnerability of the autistic brain to toxins’- as no autistic brains were part of the study.  Also ‘was associated with the amplification of autistic-like features’- was there a correlation to show this?  I did not see correlations in the Results, yet there is a lot of discussion of correlations in the Discussion.  For ‘PPA-induced autistic rats’- again there is no such thing as an ‘autistic rat’.  Autism is a clinical condition in humans.  Later ‘…and IL-17 genes observed in the brain of autistic patients seriously impaired the dendritic spine development and promoted autistic-like features’-  Was causality proven here?  Wasn’t this just an association if it was in patients?  Again ‘The overproduction of pro-inflammatory cytokines was correlated…’- I did not see any correlations in the Results.  Change ‘Remarkable’ to ‘Remarkably’ and ‘PCC, and MCC was’ to ‘PCC, and MCC were’.  What is meant by ‘Abnormal energy production is generally observed in autistic patients’.  How does action on the hypothalamus ‘justify the improvement of autistic-like features’?  GFAP may be of interest, but calling it a ’key marker’ is an overstatement.  Again for ‘brain of autistic rats, which was correlated with severe autistic-like features’- there are no autistic rats, and there were no correlations in the Results.  Later, ‘of autistic animals is evident’- do the authors mean ‘expected’ rather than ‘evident’?  Change ‘MgB protect’ to ‘MgB protects’.  Since MMP-9, … and PSD-95 deletion observed in the brains of ASD animals’- this is confusing- what was done?  Were all of these factors selectively deleted in the brains of animals? Change ‘improve clinical phenotype’ to ‘an improved clinical phenotype’.  The last sentence discusses ‘contributes to the improvement in learning and memory in autistic rats’- but the cited study is about an Alzheimer model. 

Conclusions- again there are no ‘autistic rats’, and the final sentence implies clinical claims that are unsupported by what is presented.

Author Response

RESPONSES TO REVIEWER-1

Sahin et al. have examined the effects of a novel formulation of biotin on a propionic acid-induced autism model in rats. There are findings of interest, but there are problems with the communication of a number of aspects.  Also, the understanding of the salience to autism needs significant improvement.

Comment 1. First, the Abstract discusses ‘improves autistic behaviors’ but the results in the Abstract only mention ‘improved learning and memory’- need to mention the social results in the Abstract in order to comment on autism-associated behaviors at the end. Also, can’t discuss in terms of ‘managing autism’- these findings have no bearing on the clinical management of autism at this stage.

Response: Thanks to the Reviewer. The abstract was revised as suggested

 Comment 2. Introduction, line 7 to 8 change ‘supposed’ to ‘believed’. 

Response: ‘supposed’ was changed to ‘believed’

 Comment 3. Paragraph 4 change ‘The using rates of’ to ‘The rates of use of’. 

Response:  This paragraph was rewritten.

Comment 4. The authors need to discuss the clinical salience of the PPA model for autism.  How is this recognized as an autism model? Why?  This is not done, leaving the paper only having implications for the PPA toxicity, which also just happens to have a social component. This must be addressed. 

 Response: The clinical salience of the PPA model for autism was added in introduction.

Comment 5. Also, I’m not sure why the authors need to dive into the controversial complementary treatments discussion.  This study involves a very specific treatment targeting one particular mechanism- you don’t need to make controversial statements about whether a broad range of these therapies are ‘acceptable for the management of ASD’ in general, or even the recommendation or not for Mg and B6.  Also, the authors need to discuss more about why they are using Mg biotinate in contrast other formulations such as D-Biotin.  The only thing stated is one brief sentence at the beginning of the final paragraph.

Response: The introduction part has been rearranged. The effects of Mg and biotin supplementation in ASD patients were given more detail. Information on other controversial complementary therapies was removed.

 Comment 6. Methods- need to discuss the model to introduce the reader to the appropriateness of this dose of PPA, and why the various doses of MgB were chosen.  Please rephrase ‘as clarified earlier noted’. 

Response:  The reason why 500 mg/kg subcutaneous PPA application was chosen in the study was explained in detail in the method section. Also, the doses of MgB were calculated based on 10, 100, and 500 mg biotin that is used for a 70kg adult human after adjusting doses based on metabolic body size (Shin et al., 2010; Nair and Jacob, 2016).

 Comment 7. Were the test rat and the stranger rat sex matched? 

Response: In the method part, it was stated that only male rats were included in the study.

 Comment 8. The Morris Water Maze Task description needs clarification- do the authors mean ‘functionally’ instead of ‘fictionally’?  What is meant by ‘were left in the water from each partition wall which was determined as imaginary’ and ‘it was kept on the platform and kept there for 30 seconds’ and ‘Where the platform was earlier located, the time spent by the rat was defined.’?

Response: Corrected as suggested

 Comment 9. Results- Another salience aspect- ‘severe neuronal loss and degeneration in the hippocampus’ is not at all what is seen in autism, but is found with this model in section 3.1.  Purkinje cell reduction is seen in ASD, however.  Important to link the findings specifically to ASD, and show an understanding of when it matches and when it does not match if this is to be the focus. 

Response: Hippocampal changes that can be observed in ASD and when these changes can be observed are mentioned in the discussion section.

Comment 10. Also- ‘the appearance of the Purkinje layer and cells is maintained following MgB treatment’- the phrasing sounds like the defects were maintained rather than ameliorated – please rephrase. 

Response:  It was overlooked during the final version of the article. Sentence removed from the article.

Comment 11. The text in section 3.1 and Figure 1 states the doses of MgB as 1, 100 and 500 but the rest of the paper says 10, 100, and 500- which is it? 

Response: Corrected as suggested.

 Comment 12. The increased central time in the control condition in Figure 2 seems very odd- might discuss whether there is less movement the PPA group.

Response:  Corrected as suggested.

Comment 13. Discussion- First paragraph ‘MgB could be considered an effective agent in the management/treatment of autism’- this is an extreme overstatement of what is supported by this data. The reversal of the effects of a biochemical challenge during pregnancy that might be an ASD rat model cannot lead to clinical conclusions. 

Response:  This paragraph was revised as suggested.

Comment 14. In the second and third paragraph, there is not a clear understanding of autism models- should review the mainstream models to understand what the leading models are to discuss at the beginning- and again the salience of the PPA model is not presented. 

Response: Revised as suggested both in the Introduction and Discussion.

Comment 15. Change ‘severe damages’ to ‘severe damage’. 

Response: ‘severe damages’ was changed to ‘severe damage’

 Comment 16.  Please note that the massive changes in the Purkinje cells and the neuronal loss and degeneration in the hippocampal CA1 region’ do not match well the pathology seen in autism, and there is no such thing as ‘autistic rats’. 

Response: Hippocampal changes that can be observed in ASD and when these changes can be observed are mentioned in the discussion section. Term ‘autistic rats’ changed to “PPA-induced autism-like rats” throughout the article

 Comment 17. The statement ‘The relationship between serotonin and dopamine levels and autism have not been clearly defined’ is misleading- they are rather heavily studied, and the authors should review this evidence, and hyperserotonemia is not considered a primary cause of autism by most investigators. 

Response: Studies on autism and serotonin were reviewed. The serotonin and dopamine paragraph in the discussion section has been rearranged.

Comment 18. While it is true that antipsychotics are helpful, the support for major dopaminergic dysfunction as central in ASD is very limited at best.

Response: It was stated that the data supporting the role of dopaminergic dysfunction in the etiology of ASD are limited.

 Comment 19. Later ‘One of the proposed hypotheses to explain the etiology of ASD is the imbalance between oxidants and antioxidants’- this maybe be an important factor, at least in a subset of cases, but the way this is stated sounds like it is a primary cause, which is an overstatement.  Not sure how the findings herein are ‘implying vulnerability of the autistic brain to toxins’- as no autistic brains were part of the study. 

Response: The statement re-formed as "The imbalance between oxidants and antioxidants is another factor which has been reported that it may have an effect on the emergence of ASD."

Comment 20. Also ‘was associated with the amplification of autistic-like features’- was there a correlation to show this?  I did not see correlations in the Results, yet there is a lot of discussion of correlations in the Discussion. 

Response: No correlation analysis was performed in the study. The statement re-formed as “In this study, it was determined that PPA increased oxidative stress in rats. The emergence of autism-like features in rats may be related to this increased oxidative stress.”

Comment 21. For ‘PPA-induced autistic rats’- again there is no such thing as an ‘autistic rat’.  Autism is a clinical condition in humans. 

Response:  Corrected as “PPA-induced autism-like rat model.”

Comment 22. Later ‘…and IL-17 genes observed in the brain of autistic patients seriously impaired the dendritic spine development and promoted autistic-like features’-  Was causality proven here?  Wasn’t this just an association if it was in patients? 

Response: That study was done in rats. It has been shown that autism-like behaviors occur after IL-6 increase in rats. The study was discussed in more detail in the discussion part.

Comment 23. Again ‘The overproduction of pro-inflammatory cytokines was correlated…’- I did not see any correlations in the Results. 

Response: The statement was rewritten as “It was observed that rats with more proinflammatory cytokine production had more oxidative stress and impaired cognitive function in their brains.”

Comment 24. Change ‘Remarkable’ to ‘Remarkably’ and ‘PCC, and MCC was’ to ‘PCC, and MCC were’. 

Response: ‘Remarkable’ was changed to ‘Remarkably’ and and ‘PCC, and MCC was’ was changed to ‘PCC, and MCC were’

Comment 25. What is meant by ‘Abnormal energy production is generally observed in autistic patients’.  How does action on the hypothalamus ‘justify the improvement of autistic-like features’?  GFAP may be of interest, but calling it a ’key marker’ is an overstatement. 

Response: Revised as suggested

Comment 26. Again for ‘brain of autistic rats, which was correlated with severe autistic-like features’- there are no autistic rats, and there were no correlations in the Results. 

Response: “autistic rats” was changed to “PPA-induced autism-like rat”. No correlation analysis was performed in the study. “which was correlated with severe autistic-like features” was removed.

Comment 27. Later, ‘of autistic animals is evident’- do the authors mean ‘expected’ rather than ‘evident’? 

Response: “evident” was changed to “expected.”

Comment 28. Change ‘MgB protect’ to ‘MgB protects’. 

Response:  ‘MgB protect’ was changed to ‘MgB protects.’

Comment 29. Since MMP-9, … and PSD-95 deletion observed in the brains of ASD animals’- this is confusing- what was done?  Were all of these factors selectively deleted in the brains of animals?

Response: Revised

Comment 30. Change ‘improve clinical phenotype’ to ‘an improved clinical phenotype’. 

Response: ‘improve clinical phenotype’ was changed to ‘an improved clinical phenotype’

Comment 31. The last sentence discusses ‘contributes to the improvement in learning and memory in autistic rats’- but the cited study is about an Alzheimer model. 

Response: It was rearranged as “Streptozotocin-Induced Sporadic Alzheimer’s Model rats”

Comment 32. Conclusions- again there are no ‘autistic rats’, and the final sentence implies clinical claims that are unsupported by what is presented.

Response:  “autistic rats” was changed as “PPA-induced autism-like rat”

Reviewer 2 Report

The paper "Therapeutic Effects of a Novel Form of Biotin on Propionic Acid-Induced Autistic Features in Rats" describes the effects of a novel Magnesium biotinate complex on autistic features in rats. This paper is not clearly presented and executed. Such sections as "Introduction" and "Discussion" are written simplistically. In the "Material and Methods" section authors haven’t described cell counting mechanisms and areas in the cerebellum where cell count took place in enough detail. Also, in this section, the authors have mentioned that homogenate and supernatant were frozen and stored at -80 degrees Celsius which, according to my experience, severely affects the activity of enzymes such as GPX and SOD.

Author Response

RESPONSES TO REVIEWER-2

The paper "Therapeutic Effects of a Novel Form of Biotin on Propionic Acid-Induced Autistic Features in Rats" describes the effects of a novel Magnesium biotinate complex on autistic features in rats. This paper is not clearly presented and executed. Such sections as "Introduction" and "Discussion" are written simplistically. In the "Material and Methods" section authors haven’t described cell counting mechanisms and areas in the cerebellum where cell count took place in enough detail. Also, in this section, the authors have mentioned that homogenate and supernatant were frozen and stored at -80 degrees Celsius which, according to my experience, severely affects the activity of enzymes such as GPX and SOD.

           Response: Thank to the Reviewer. The introduction and discussion parts were rewritten. In addition, Tissues were analyzed immediately after homogenization. The sentence was misspelled.  Corrected as suggested.

Round 2

Reviewer 1 Report

Re: nutrients-1552763

Sahin et al have resubmitted a paper on the effects of a novel formulation of biotin on a propionic acid-induced autism model in rats. There are findings of interest, but sum issues remain.

Introduction, new text- ‘yielded similar data in ASD patients’ should be ‘yielded similar findings to what is seen in ASD patients’.  Also, ‘animal research using propionic acid helps autism-like rat models for ASD research’- please rephrase- this could be misinterpreted as saying propionic acid reverse ASD-like behaviors in rats.

Methods- A couple of minor issues were not addressed: For the social task please rephrase ‘as clarified earlier noted’.  Were the test rat and the stranger rat sex matched?  For the Morris Water Maze Task please clarify ‘Where the platform was earlier located, the time spent by the rat was defined.’?

Results- A salience aspect- ‘severe neuronal loss and degeneration in the hippocampus’ is not at all what is seen in autism, but is found with this model in section 3.3.  Purkinje cell reduction is seen in ASD, however.  Important to link the findings specifically to ASD clinically, and show an understanding of when it matches and when it does not match if this is to be the focus.  Figure 1C is no longer called out in the text.  The increased central time in the control condition in Figure 2 seems very odd- might discuss whether there is less movement the PPA group.

Discussion- In the second and third paragraph, there is not a clear understanding of autism models- should review the mainstream models to understand what the leading models are to discuss at the beginning, THEN mention the present models.  Of the models mentioned, on the valproate model is commonly used, and ODDLY it is listed twice (‘sodium valproate, valproic acid’).   Please note that the massive changes in the Purkinje cells and the neuronal loss and degeneration in the hippocampal CA1 region do not match well the pathology seen in clinical autism.  Change ‘bio-tin’ to ‘biotin’.  GFAP may be of interest, but calling it a ’critical marker’ may be an overstatement.

Conclusions- the final sentence should include a caveat that this may be applicable only in some models of autism, since this is only assessed in the PPA model and can’t be presumed to have effects across autism.

Author Response

REVIEWER 1

Sahin et al have resubmitted a paper on the effects of a novel formulation of biotin on a propionic acid-induced autism model in rats. There are findings of interest, but sum issues remain.

 Comment 1. Introduction, new text- ‘yielded similar data in ASD patients’ should be ‘yielded similar findings to what is seen in ASD patients’. 

Response: Revised as suggested.

Comment 2. Also, ‘animal research using propionic acid helps autism-like rat models for ASD research’- please rephrase- this could be misinterpreted as saying propionic acid reverse ASD-like behaviors in rats.

Response: Revised as suggested

Comment 3. Methods- A couple of minor issues were not addressed: For the social task please rephrase ‘as clarified earlier noted’. 

Response: Revised as suggested

Comment 4. Were the test rat and the stranger rat sex matched? 

Response: Revised as suggested

Comment 5.  For the Morris Water Maze Task please clarify ‘Where the platform was earlier located, the time spent by the rat was defined.’?

Response: Revised as suggested

Comment 6. Results- A salience aspect- ‘severe neuronal loss and degeneration in the hippocampus’ is not at all what is seen in autism, but is found with this model in section 3.3.  Purkinje cell reduction is seen in ASD, however.  Important to link the findings specifically to ASD clinically, and show an understanding of when it matches and when it does not match if this is to be the focus. 

Response: Thanks to the Reviewer. ‘severe neuronal loss and degeneration in the hippocampus’ was removed. In addition, in the discussion section, more detailed information was given about the relationship between autism and the hippocampus.

Comment 7. Figure 1C is no longer called out in the text. 

Response: “Figure 1C” has been added to the text again.

Comment 8. The increased central time in the control condition in Figure 2 seems very odd- might discuss whether there is less movement the PPA group.

Response: Added into Discussion Section

Comment 9. Discussion- In the second and third paragraph, there is not a clear understanding of autism models- should review the mainstream models to understand what the leading models are to discuss at the beginning, THEN mention the present models.  Of the models mentioned, on the valproate model is commonly used, and ODDLY it is listed twice (‘sodium valproate, valproic acid’).  Response: In the discussion part, general information about the ASD animal models was given, then the information was given about the most commonly used VPA model. Then the present model was mentioned. “sodium valproate” was deleted.

Comment 10. Please note that the massive changes in the Purkinje cells and the neuronal loss and degeneration in the hippocampal CA1 region do not match well the pathology seen in clinical autism. 

Response: It was stated that these changes were not consistent and there was no correlation between clinical features and these alterations.

Comment 11. Change ‘bio-tin’ to ‘biotin’. 

Response: ‘bio-tin’ was changed to ‘biotin’

Comment 12. GFAP may be of interest, but calling it a ’critical marker’ may be an overstatement.

Response: The comment was rearranged as “Thus, it is stated that GFAP may be used as a marker for brain damage as observed in ASD patients.”

Comment 13. Conclusions- the final sentence should include a caveat that this may be applicable only in some models of autism, since this is only assessed in the PPA model and can’t be presumed to have effects across autism.

Response: It was stated that the study was conducted only in PPA-induced autism-like rats and further studies are needed to determine the effect of MgB in patients with autism.

Reviewer 2 Report

This paper is poorly designed and I don't believe it to be highly important for the scientific community, but it was slightly improved by the authors.

Author Response

Response: Thanks to the Reviewer. The paper was improved.